# Genetic Parameters of Honey Bee Colonies Traits in a Canadian Selection Program

**DOI:** 10.3390/insects11090587

**Published:** 2020-09-01

**Authors:** Ségolène Maucourt, Frédéric Fortin, Claude Robert, Pierre Giovenazzo

**Affiliations:** 1Department of Biology, Vachon Pavillon, Université Laval, Québec, QC G1V 0A6, Canada; Pierre.giovenazzo@bio.ulaval.ca; 2Centre de Développement du porc du Québec, 450-2590 Boulevard Laurier, Québec, QC G1V 4M6, Canada; ffortin@cdpq.ca; 3Department of Animal Science, Institut sur la Nutrition et les Aliments Fonctionnels, Université Laval, Québec, QC G1V 0A6, Canada; Claude.Robert@fsaa.ulaval.ca

**Keywords:** heritability, honey bee, selection, genetic correlation, breeding program

## Abstract

**Simple Summary:**

Genetic selection is still little applied to honey bees (*Apis mellifera*), whose complex genetic and reproductive characteristics are a challenge to model statistically. The aim of this project was to determine the genetic parameters of several traits important for Canadian beekeepers with a view to establishing a breeding program in a northern context. Our results show that the five traits studied (i.e., honey production, spring development, winter consumption, hygienic behavior and *Varroa destructor* infestation) are all heritable. Furthermore, the genetic correlations between these traits are all positive or null, which means that these traits can be selected simultaneously in a honey bees genetic selection program. Our results are instrumental to the development of a selection index that can be used to improve the capacity of honey bees to thrive in northern conditions.

**Abstract:**

Genetic selection has led to spectacular advances in animal production in many domestic species. However, it is still little applied to honey bees (*Apis mellifera*), whose complex genetic and reproductive characteristics are a challenge to model statistically. Advances in informatics now enable creation of a statistical model consistent with honey bee genetics, and, consequently, genetic selection for this species. The aim of this project was to determine the genetic parameters of several traits important for Canadian beekeepers with a view to establishing a breeding program in a northern context. Our results show that the five traits measured (*Varroa destructor* infestation, spring development, honey production, winter consumption, and hygienic behavior) are heritable. Thus, the rate of *V. destructor* infestation has a high heritability (*h*^2^ = 0.44 ± 0.56), spring development and honey production have a medium heritability (respectively, *h*^2^ = 0.30 ± 0.14 and *h*^2^ = 0.20 ± 0.13), and winter consumption and hygienic behavior have a low heritability (respectively, *h*^2^ = 0.11 ± 0.09 and *h*^2^ = 0.18 ± 0.13). Furthermore, the genetic correlations between these traits are all positive or null, except between hygienic behavior and *V. destructor* infestation level. These genetic parameters will be instrumental to the development of a selection index that will be used to improve the capacity of honey bees to thrive in northern conditions.

## 1. Introduction

Genetic selection has led to spectacular progress in the production of many animal species [1,2,3]. Indeed, in the last thirty years, the integration of quantitative genetics into selection methods has allowed to refine breeding choices based on statistical observations and to improve animal production both in terms of quality and quantity [4,5]. In honey bees, such selection has long been impossible because of specific aspects of their reproduction, genetic architecture, sensitivity to inbreeding, and social nature, which often define them as super organisms [6,7]. Indeed, the queen is the primarily element responsible for the overall genetics of each colony. However, the result of phenotypic measurement taken on the colony corresponding to the genetic contribution of the two generations present in the colony: The queen, and her daughters, who are the workers [8,9]. The queen does not contribute directly to the performance traits, but influences them through the contribution of her genes to her workers or through her egg production, whereas workers influence performance traits directly through the different activities they perform within the colony, for example, collecting nectar or cleaning cells [10]. All these characteristics of genetic model specific to the honey bee have long been difficult to model statistically by computer scientists and statistician in comparison with other animal productions. Notably because of the presence of two distinct groups of father (Figure 1) involved in the expression of the phenotype of a colony: The fathers-of-queen group and the fathers-of-workers group [11].

As a result, there are still very few sustainable genetic selection programs for honey bees worldwide [10,12]. To our knowledge, the only such program is the European “Beebreed”, which has been underway since the early 1990s [13]. This scheme performs genetic evaluations of colonies using the Best Linear Unbiased Prediction (BLUP)-Animal Model, which has been adapted to the reproductive and genetic characteristics of honey bees [14]. Genetic evaluations obtained with BLUP-Animal Model have allowed genetic improvement of several traits in honey bees, such as honey production or gentleness [14,15].

The BLUP-Animal Model is a promising tool for integration in a Canadian genetic selection program to improve honey bee production, health status and hardiness, which are greatly prized by Canadian beekeepers. Indeed, while some Canadian beekeeping operations are in temperate regions, most are in cold climates, where the honey flow season is short (June–August) and the wintering time is long (October–April) [16,17,18,19]. This has an important impact on beekeeping management and requires the use of honey bees adapted to these conditions [20,21]. The Canadian beekeeping industry needs hardy honey bee lines able to survive the long winter months with rapid population growth in spring [22]. In addition, since 2007, Canada has suffered abnormal and significant colony losses in winter, averaging 26% per year [23,24]. Several factors affecting the health of honey bee colonies; like parasitism by *Varroa destructor* and *Nosema* spp., shortage of flower resources, pesticide exposure, or stresses associated with pollination services; are causing important winter colonies losses in Canada [23,25,26]. To compensate for these losses, the Canadian beekeeping industry imports package bees and queens from abroad, mainly from Australia, New-Zealand, Chile, and the USA (California or Hawaii). Unfortunately, these imports are unreliable solutions, because they are dependent of export reliability and the imported honey bees are not adapted to the northern climate and Canadian beekeeping industry practices [27]. This leads to poor winter survival rates, increased queen mortality, and reduced overall colony productivity [20,28,29]. Genetic improvement of the health, productivity, and rusticity of local honey bee stocks through a genetic selection program would therefore be a sustainable solution that would reduce honey bee imports and help maintain Canadian honey bee industry self-sufficient.

Before colonies can be selected from breeding values, it is important to estimate the genetic parameters of the traits of interest for selection (i.e., variance population, heritability, and genetic correlation) to ensure efficacy of the genetic program [30]. With the BLUP-Animal Model, genetic evaluation is performed on animals by integrating their genealogical information (i.e., colony pedigree) and the effects of the specific environment in which the animal evolves (e.g., in honey bees: The apiary, the year, and the various health treatments). This can be defined using the following mixed linear model equation:P=G+N+E
where *P* corresponds to the phenotypic value of the animal, *G* to the genetic value of the individual (random effects), *N* to the identified effects of the environment (fixed or random effects depending on the choice of modeling) and *E* to the residual effects (random effects). Additionally,
G=A+I
where *A* is the additive value and *I* represents the interaction effects (i.e., dominance and epistasis of gene effects, these effects are generally ignored in the genetic study of quantitative traits and will be ignored in this study). Thus,
P=A+I+N+E

Resolving this equation not only predicts the genetic value of an animal for selection but also indirectly allows the phenotypic variance of the population to be broken down to obtain the variance of the genetic origin of a trait in a population. We can then estimate the genetic parameters of traits such as heritability, or genetic correlations between traits [31,32].

The heritability genetic parameter is represented by an index (0 < *h*^2^ < 1) that evaluates the proportion of the phenotypic variation of a trait that is of genetic origin in a specific population (i.e., additive variance, *A* in the equation). This index is used to predict the promptness of response of a selected trait. A higher value of the index is associated to higher transmission of the trait from one generation to the next, thus, the response to selection will be faster [4,33]. Another important genetic parameter is the genetic correlation index (−1 < *r* < 1). This index measures the intensity of the link between two quantitative traits [4]. Variation of the traits over time can be similar (*r* > 0), opposite (*r* < 0) or independent (*r* = 0) [34]. Most often, breeders are interested in several traits simultaneously, therefore, success of a selection program relies on good knowledge of the genetic correlations between the selected performance traits. This will result in an increase in the global performance of the selected lines [34].

To our knowledge, little information is available in the technical or scientific literature on the use of various genetic parameters in North American honey bee breeding programs. The aim of our study is to estimate the genetic parameters (heritability and genetic correlations) of several valuable traits of interest to our beekeeping industry (*Varroa destructor* infestation, spring development, honey production, winter consumption, and hygienic behavior).

## 2. Materials and Methods

### 2.1. Biological Material

Our study was conducted in honey bee colonies at the Centre de recherche en sciences animales de Deschambault (CRSAD) Québec, Canada (N46°40.27′, W10°71.50′). The CRSAD has housed a breeding program based on phenotypes since 2010. The selection program started with 26 mother colonies. The queens of eleven of these colonies were from local breeders in Quebec or elsewhere in Canada with European-derived stock, as well as 15 colonies of Buckfast lines imported from Denmark (Buckfast Denmark, https://buckfast.dk/index.php/en/). After evaluating the performance of these colonies, 7 were selected for queen production and 10 for drone production to together produce 100 queens (15 queens per maternal lineage) who would constitute the first generation of the breeding program (year 2011): The entire resulting breeding program is therefore based on this common ancestral stock.

### 2.2. Pedigree Determination

The selected traits used for selective breeding (e.g., kg honey produced and percentage of hygienic behavior of honey bees) were measured at the colony level (i.e., experimental unit). Colonies kept for breeding (queen-producing colonies and drone-producing colonies) were therefore selected based on their phenotypic performance only, hence, the males produced by entire group of drone-producing colonies are not related. Our pedigree database has information on the genealogical background of all the colonies in our breeding program. For each queen in the pedigree database, we identified her mother, the entire group of fathers involved in the fertilization of her mother, the entire group of fathers involved in her fertilization, her sisters and her daughters (if she was selected for breeding) [35]. A unique identification number was assigned to each queen and her colony [36], so that performances are associated to the queen and her workers. This identification number is used in the performance and the pedigree databases allowing linkage between them (see Section 2.3). This unique queen identifier comprises 3 elements: year of birth of the queen (2 last digits), breeder code (1 letter) and queen number within the studbook of the breeder (3 digits).

In total, our pedigree database contains the genealogical information of 604 colonies across 8 generations from 2010 to 2017 (performance measures from 2011 to 2018). Each generation includes about a hundred colonies and is composed of 7 to 14 different queen-producing colonies, depending on the year. From each of these selected colonies we produced 10 to 12 sister-queens at the start of the generation (Table 1).

Each year, about 20 colonies with the best overall performance were selected for breeding. Of about 20 colonies selected, between 7 to 14 were used for queen production, the mothers-of-queens, and the other 10 were used for drone production, the fathers-of-workers (Figure 2). The choice of breeder colonies was based only on their performance and did not consider the existing kinship links between them.

Young queens were produced from the selected mother lines (10 to 12 per generation) using the Doolittle queen rearing method [37]. One day before emerging, queen cells were placed in mating nucs (dimensions: 12 1/2″ × 7 7/8 × 8 5/8″; each containing 2 full brood frames with the young adherent honey bees, an empty frame with drawn cells, and a frame with honey and pollen) in a mating yard near the drone-producing colonies (1200 m). A drone frame (Propolis-etc…, Saint-Pie, QC, Canada; PL-1900) was placed in the center of the brood chamber of each drone-producing colony selected (i.e., fathers-of-workers) to ensure flooding of selected drones in the area. This method has been used in other breeding plans that confirm 83 to 93% of mating is performed by the selected males [38]. This mating design is less binding than artificial insemination or geographic isolation and provides higher selection pressure than open mating (i.e., no mating control) [39]. In addition, we have identified a male congregation area adjacent to our drone-producing colonies located 1.6 km from our mating yard (Figure 3) [40]. The male pedigree is not included in our statistical analyses, but it is included in our pedigree databases. Drone-producing colonies are selected as well as queen-producing colonies. Unfortunately, there is no guarantee that our queens are 100% fertilized by these selected males but we are confident that most of our queens are fertilized by them [38].

All queens were identified by marking the back of their thorax with a queen marking pen (Propolis-etc…, Saint-Pie, QC, Canada; MP-1103 to MP-1104) and clipping half of the right wing to prevent them from swarming. Colonies were inspected regularly (every 15 days from early June to late August) to destroy royal cells. However, if a queen swarmed or died before or during the performance measurement period, the colony was excluded from the breeding program and the performance measured after the event was no longer included in the databases. These precautions ensured the pedigree of the queens and the validity of data [12,41].

The generation interval of our breeding program is two years. Each year, colonies are evaluated on their performance traits (one full beekeeping season: Summer–Fall–Winter–Spring) and, in July of the second year, best performing colonies are selected for the next generation of queens/lines (Figure 2). The queens produced from the selected colonies are introduced in double nuclei Langstroth 4 frames (Propolis-etc…, Saint-Pie, QC, Canada; NU-2002), with 2 full brood frames each with its young adherent honey bees, an empty frame with drawn cells and a frame with honey and pollen. Prior to wintering, each nucleus is fed 10 L of sucrose–water solution (2:1) using a double nuclei feeder with floaters (Propolis-etc…, Saint-Pie, QC, Canada; FE-1700) and is treated against *Varroa destructor*. The varroa treatment alternates every second year between Thymovar^®^ and Apivar^®^. In mid-November, all double nuclei are overwintered in an environmentally controlled room (4 ± 1 °C and 40–50% RH). The following spring (early May), the queens and their colonies are transferred in Langstroth 10-frame hives (Propolis-etc…, Saint-Pie, QC, Canada; HC-1100) and equally distributed between 4 different apiaries, so that sister-queens of all the lines are present in all apiaries [36]. These apiaries are situated within a radius of 20 km of our research center (CRSAD) and at least 5 km away from each other in similar agricultural environment with the same potential honey production. These colonies are managed for honey production and their performance is evaluated until the following year in order to include a complete beekeeping season.

Each year, problematic colonies (lost/dead queen, drone laying queen, dead colony, swarming) were removed from the selection program and all data from these colonies was removed from the pedigree and performance databases. Of all the colonies produced at each generation, 50 to 70 colonies were integrated in our databases each year from 2010 to 2018.

The drone-producing colonies (fathers-of-queens or fathers-of-workers) were selected according to their global performance, but the kinship relationship between these selected colonies were not considered, thus statistical modelling of the pedigree of males was not considered in our model. Integrating the male pedigree into our statistical model would have altered the accuracy of traits heritability estimations. The statistical modelling of our selection scheme is therefore was made only on the female side.

### 2.3. Traits Measured to Evaluate Colony Performance

The traits studied were chosen taking into account the requirements and challenges reported by the Canadian beekeeping industry [42,43,44]. The traits measured are productivity, measured by honey production; health, specifically hygienic behavior and *V. destructor* infestation level; hardiness, as reflected in winter weight loss and spring development.

#### 2.3.1. Productivity of Honey Production

Colonies are provided with honey supers, which are placed above a queen excluder. Each colony has at least two supers during honey flow, each with frames of drawn comb. Honey weight gain is obtained by weighing honey supers when added and removed from a colony or by placing the entire colony (brood chamber and honey supers) on a platform scale (CAS-USA, East-Rutherford, NY, USA; CAS CI-2001BS).

#### 2.3.2. Health

Hygienic behavior: To perform the test, the colony must be opened, and a comb is selected that contains a solid patch of sealed worker brood at the pupal stage, with pupae having pink or purple eye color. Two PVC tubes (2″ inside diameter) are pressed down to the midrib of the comb. The number of empty cells (“misses”) are counted in each tube. Liquid nitrogen is then applied, 300 mL/tube, to “freeze kill” the brood. Frames are marked and returned to the colony. After 24 h, the number of cells that remain capped or partially removed are counted. The total number of cells removed are counted and yield a percentage of hygienic behavior [45,46]. This measure was taken before the first honey flow at the end of May or during June between 2011 and 2015 and then after the summer honey flow during August in 2016 and 2017. In 2015, the hygienic behavior test was not performed on all colonies of our honey bee population.

*V. destructor* infestation level was assessed by the natural mite fall method using sticky boards placed on the bottom boards of hives [47,48]. Mites that had fallen on sticky boards (5–7 days) were counted to obtain a daily mite drop value. Infestation rates were measured in May and in September. Comparisons between colonies were based on the September infestation rates in relation to the initial May infection rates every year except 2015, 2016, and 2017 (no data).

#### 2.3.3. Hardiness

Winter weight loss: This reflects the consumption of sucrose syrup during winter. It is calculated as the difference in colony weight before vs. after wintering (November vs. April). This measure was taken during the second wintering, when colonies were wintered in one brood chamber (not for the 4 frame nucs winter the first year). Colonies were weighed individually using a platform scale (CAS-USA, East-Rutherford, NY, USA; CAS CI-2001 BS).

Spring development: Colony strength was evaluated by measuring the area occupied by immature worker honey bees (eggs + larvae + capped brood) in colonies early June. This was done by measuring width and length of the brood surface area on each side of every brood frame. The rectangular surface obtained was multiplied by 0.8 to compensate for the elliptic shape of the brood pattern. These values were added to calculate the total brood surface in each colony. A factor of 25 worker cells per 6.25 cm^2^ (i.e., a square inch) was used to convert the area to obtain the number of immature workers honey bees [49,50].

All relevant information on the performance measurements was compiled using another database, performance database. This database contains the performance measurements associated with each colony, the apiary, the year in which the measurements were taken and the unique identifiers of each queen belonging to the breeding program (defined in Section 2.2, these identifiers are also present in the pedigree database).

### 2.4. Statistical Analysis

Statistical tests were conducted using SAS software (ver. 9.4, SAS Institute Inc., Cary, NC, USA) at the 0.05 level of significance. The databases containing colony pedigree and associated performance were merged using the SAS MERGE procedure. Data were previously sorted according to the queen’s unique identifier with the SAS SORT procedure. All dependent variables were tested for normality using the Skewness and Kurtosis test, and a Box-Cox power transformation was used when necessary to meet the normality assumptions of the model (only data of hygienic behavior trait and *V. destructor* infestation level trait have been transformed using the logarithm function). A kinship relationship matrix was established with the SAS INBREED procedure to obtain the relationship coefficients between all colonies of the breeding program since 2010. Multiple father ids are recorded in the mating of a queen, so this information was not included in our analysis. Variance components of phenotype traits were estimated with the Residual Maximum Likelihood (REML) Animal Model using the SAS MIXED procedure, with the LDATA option. This allows the integration of relation coefficients and their data set within the genetic relationship matrix. The univariate model used for our data analysis is as follow:$$P_{ijk}=µ+year_{i}+apiary_{j}+year\times apiary_{\left(ij\right)}+colony_{k}+e_{ijk}$$
where: *P_ijk_* = performance trait (i.e., hygienic behavior, honey production,…) for colony k; µ = general mean of the population for this phenotype; *year_i_* = fixed effect of the year; *apiary_j_* = fixed effect of the apiary; *year* × *apiary_(ij)_* = fixed effect of interaction between year and apiary; *colony_k_* = random genetic direct effect of colony k assumed distributed ~N(0, Aσ2a), with A equal to the relationship matrix constructed using SAS INBRED procedure; and *e_ijk_* = residual associated with the measurement. Once the variance analysis was calculated with the model described above, the heritability index for each trait was then obtained by the following formula:$$h^{2}=\frac{V_{Additive}}{V_{Residual}+V_{Additive}}\;=\;\frac{V_{Additive}}{V_{Phenotype}}$$

SAS software does not deliver standard errors for heritability directly in the outputs, so they were estimated from the variances of each trait with the following formulas within an Excel spreadsheet [51]:$$Var\;{\left(h\right)}^{2}={\left(\frac{V_{Additive}}{V_{Phenotype\;}}\right)}^{2}\left(\frac{var(V_{Additive})}{V_{Additive}{}^{2}}+\frac{var(V_{Phenotype})}{V_{Phenotype}{}^{2}}-2\frac{cov(V_{Additive},V_{phenotype})}{V_{Addtitive}V_{Phenotype}}\right)$$
and,
$$SE\left(h^{2}\right)=\sqrt{Var\left(h^{2}\right)}.$$

Genetic correlations have been estimated from the Pearson correlations between breeding values from the univariate models for each trait with positive heritability using the SAS CORR procedure. Bivariate or multivariate models have not been used for these estimations.

## 3. Results

Distributions of phenotypic data measured from 2010 to 2018 (Figure 4) showed a high variability within the colonies of the breeding program for all the traits considered in this study. Actually, honey production (Figure 4A, median = 49, mean = 51.2 ± 33.4); hygienic behavior (Figure 4B, median = 81.6, mean = 75.9 ± 21.7); varroa destructor infestation level (Figure 4C, median = 8, mean = 19 ± 32); spring development (Figure 4D, median = 21,740, mean = 21,438 ± 8951); and winter weight loss (Figure 4E, median = 10.4, mean = 10.5 ± 2.6) showed there are a large phenotypic variation in the colonies of breeding program, and thus, a potential improvement of these traits by selection.

Estimates of heritability (*h*^2^) for traits measured between 2010 and 2018 ranged from 0.11 to 0.44 (Table 2). All the traits were therefore heritable and can be genetically selected. Heritability of *V. destructor* infestation level trait showed high heritability (*h*^2^ = 0.44 ± 0.56). Heritability values for spring development and honey production showed medium heritability (respectively, *h*^2^ = 0.30 ± 0.14 and *h*^2^ = 0.20 ± 0.13) and the winter weight loss and hygienic behavior showed low heritability (respectively, *h*^2^ = 0.11 ± 0.09 and *h*^2^ = 0.18 ± 0.13) [52].

The genetic correlations between traits (Figure 5) show that hygienic behavior was positively correlated with honey production (*r* = 0.09, *n* = 604, *p* < 0.05), spring development (*r* = 0.12, *n* = 604, *p* < 0.01) and winter weight loss (*r* = 0.16, *n* = 437, *p* < 0.001). This is also the case between spring development and winter weight loss (*r* = 0.25, *n* = 437, *p* < 0.001) and honey production (*r* = 0.50, *n* = 604, *p* < 0.001). Hygienic behavior was negatively correlated with *V. destructor* infestation level (*r* = −0.42, *n* = 224, *p* < 0.001). Honey production was not correlated to *V. destructor* infestation level (*r* = 0.05, *n* = 224, *p* = 0.4315) and winter weight loss (*r* = 0.04, *n* = 437, *p* = 0.3871), just as the *V. destructor* infestation level was not correlated to winter weight loss (*r* = 0.01, *n* = 224, *p* = 0.9203) and spring development (*r* = 0.01, *n* = 224, *p* = 0.9003).

## 4. Discussion

The aim of this project was to study the genetic parameters (variance population, heritability, and genetic correlation) of several beekeeping traits that are important for Canadian honey bee breeding programs. To achieve this aim, we first studied the distribution of data for each trait and then, we measured the variance additive value of these traits in CRSAD breeding program colonies using the REML approach. The variance additive value for each trait was then used to calculate the heritability index for each trait, as shown in the flow chart in Figure 6. If the trait is heritable (*h*^2^ > 0), the genetic correlations between these traits are estimated to determine their degree of genetic relationship in the population. Information on the genetic parameters of beekeeping traits as well as their associated economic values will then be taken into consideration for the establishment of the selection index, according to the relative value will be attributed to them by the breeder. Otherwise, if the trait is not heritable (*h*^2^ = 0), it cannot be improved by selection using the current model and data. However, other mechanisms can be considered to enhance this trait, for example, improving colony nutrition, colony management, and apiary site management or choice [4,34].

In this study, the genetic parameters of the selected traits allowed us to affirm it is possible to improve zootechnical performance for these traits through a selection program. Indeed, all traits studied showed genetic variance in the population, meaning that these traits have the potential to be genetically improved. Also, all these traits have low to high heritability, which means that genes associated with these traits are transmitted from one generation to the next and that the trait will improve with selection [4,33]. In addition, a higher value of the heritability indicates a greater link between the genotype and phenotype, and stronger selective pressure can be applied. In other words, the higher the heritability value, the faster the improvement in the performance trait [33,34]. As a result, we can predict the effectiveness of genetic selection for the traits studied.

Thus, spring development and honey production traits will improve relatively rapidly (with respectively *h*^2^ = 0.30 ± 0.14 and *h*^2^ = 0.20 ± 0.13, considered medium). Heritability for honey production and spring development were similar to the estimated heritability indexes for these same traits in other honey bee populations in Europe, with heritability ranging from 0.23 to 0.45 for the honey production trait [53,54,55,56,57,58] and a heritability value that varies from 0.33 to 0.35 for the spring development trait [54,59]. These similarities strengthen the reliability of our results and our statistical model, especially since the honey bee population studied comes largely from stock imported from Europe (see Section 2.1). Indeed, a heritability index is specific to a population because it depends on the genetic constitution of the colonies that form this population. The closer populations are genetically for a trait, the more likely they are to have a similar heritability for this trait [4]. Then, hygienic behavior trait will improve more slowly than other traits studied (*h*^2^ = 0.18 ± 0.13, considered as low). However, even if our heritability value for the hygienic behavior trait is lower than that estimated by Facchini et al. [60] (*h*^2^ = 0.37 ± 0.25), by Guarna et al. [61] (*h*^2^ = 0.57) or by the USDA Honey Bee Lab at Baton-Rouge, Lousiana (*h*^2^ = 0.65 ± 0.61) [62], this previous study showed that hygienic behavior trait could be improve in a population. The Baton-Rouge research team has been selecting varroa mite-resistant honey bee stock for several years, and this could have modified the frequency of alleles in their population and increased the value of the heritability value for hygienic behavior [4]. Finally, *V. destructor* infestation level trait will improve quickly (with *h*^2^ = 0.44 ± 0.56, considered high) and winter weight loss will improve slowly (with respectively, *h*^2^ = 0.11 ± 0.09). Unfortunately, we were unable to compare the heritability for these two traits because, to our knowledge, there are no heritability estimates using variance analysis and the same methodology for data collection of phenotypic measurement.

Integrating the male pedigree into our statistical model would have altered the accuracy of traits heritability estimations. The statistical modelling of our selection scheme is therefore based on the female side, which could have underestimated some results. Furthermore, three years of missing data for the *V. destructor* infestation level (2015 to 2017) and hygienic behavior (2015) resulted in large standard errors. In the future, we plan to continue evaluation of the performance traits using a measurement rather than a classification system (i.e., traits are scored on a previously established scale). This yields phenotypic measurements with good accuracy and allows a robust estimation of the genetic value [34,36,63]. But also, to increase the accuracy of our genetic estimates by increasing the number of colonies in our program to include at least 100 colonies per year. We will also modify the selection scheme, by including more details on drone’s pedigree. Colonies selected from breeding value to produce drones for breeding will all have sister queens (i.e., sister drone-producing colonies), so the males used in our selection program will all be cousins, and thus we will be able to retrieve the information on the male pedigree in our statistical model, as proposed by Bienefeld and his collaborators [14].

Estimating genetic correlations between heritable traits allows to anticipate whether selection on one trait will have collateral impacts on other traits which needs to be considered in multi traits selection. In our study, some traits were positively correlated, meaning that both traits will improve when selecting only one trait, and some traits had no correlation, meaning that the traits are genetically independent and therefore the selection of one will not have no impact on the other. Several of the genetic correlations calculated are consistent with the literature. For example, spring development was genetically correlated with honey production (*r* = 0.50, *n* = 604, *p* < 0.001), which seems obvious because it has been shown that the larger the honey bee population in spring, the greater the number of bees available for honey harvesting and the greater the honey production [64]. Also, it has been demonstrated that winter consumption of sugar syrup by a colony is higher in those with a large honey bee population [22]; this biological link perfectly explains the significant positive genetic correlation between spring development and winter consumption proven in this study (*r* = 0.25, *n* = 437, *p* < 0.001).

One negative genetic correlation between the different traits was estimated, between hygienic behavior and *V. destructor* infestation level (*r* = −0.42, *n* = 224, *p* < 0.001). However, this is a desirable result since we seek to obtain colonies with the highest levels of hygienic behavior and the lowest varroa mite population [48]. This negative correlation confirms that the selection of hygienic behavior will reduce the *V. destructor* infestation level. Given these results, future development should look at multivariate models for the prediction of breeding values to integrate the genetic correlations between traits.

## 5. Conclusions

The objective of this study was to estimate genetic parameters with a view to improve their zootechnical performance through a genetic selection program. Traits were chosen for this study based on their importance in beekeeping in northern climates, as Canadian beekeepers increasingly favor a honey bee with high productivity (high honey production), good hardiness (rapid spring development and low winter weight loss) and that is also healthy (high hygienic behavior and low *V. destructor* infestation level). In conclusion, we have shown that the five traits studied are heritable and independent of each other or favorably correlated. Therefore, selection for these traits will lead to genetic improvement and for optimize response should be performed simultaneously with a selection index in the context of a selection program. In the future, the statistical modelling and kinship relationship matrix constructed in this study will allow us to pursue an effective breeding program, to continue the development of the genetic evaluation program and to study the genetic parameters of other important traits such as gentleness, suppressed mite reproduction [65], and pollen or propolis production. This will help ensure the sustainability and productivity of the Canadian beekeeping industry.

## Figures and Tables

**Figure 1 insects-11-00587-f001:**
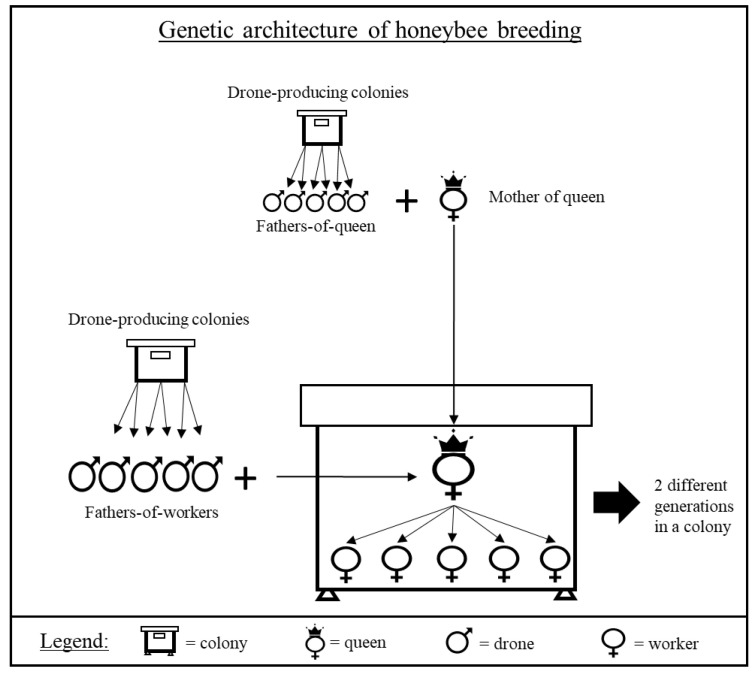
Genetic architecture of honey bee breeding. The queen of the colony is the product of mating between the mother of the queen and drones from the drone-producing colonies selected: The fathers-of-queen group. The workers of the colony are the product of mating between the queen and drones from other drone-producing colonies selected: the fathers-of-workers group.

**Figure 2 insects-11-00587-f002:**
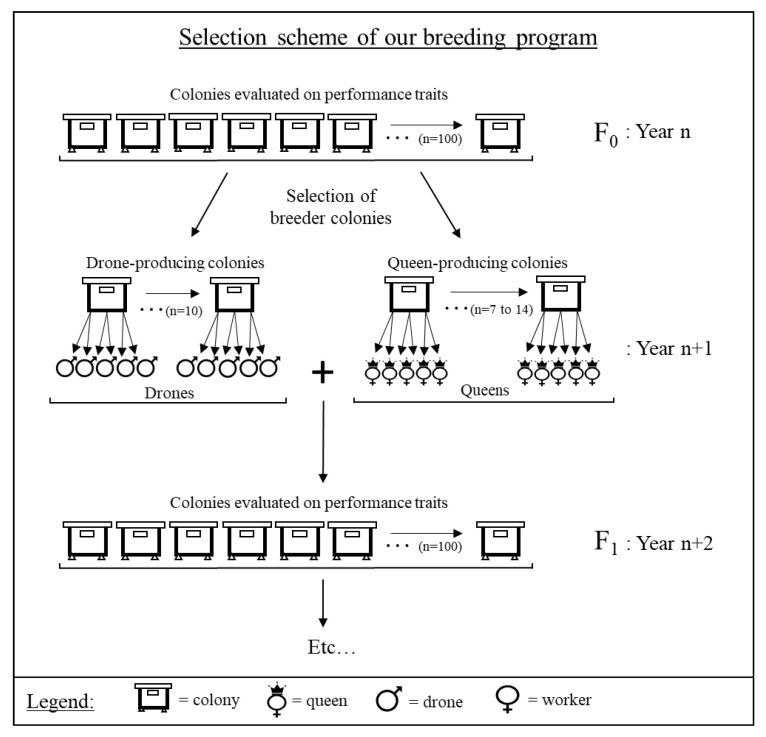
Selection scheme of our breeding program. Colonies of the F_0_ generation were evaluated on their performance traits (year n). About twenty breeder colonies were selected from this F_0_ generation (year n+1): between 7 to 14 colonies were selected to produce drones, and 10 were selected to produce queens. The queens and drones produced were mated together to constitute the F_1_ generation, which was evaluated on performance traits in year n+2.

**Figure 3 insects-11-00587-f003:**
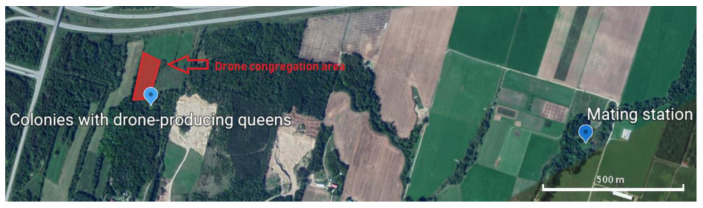
Location of drone congregation area (red zone) and mating yard.

**Figure 4 insects-11-00587-f004:**
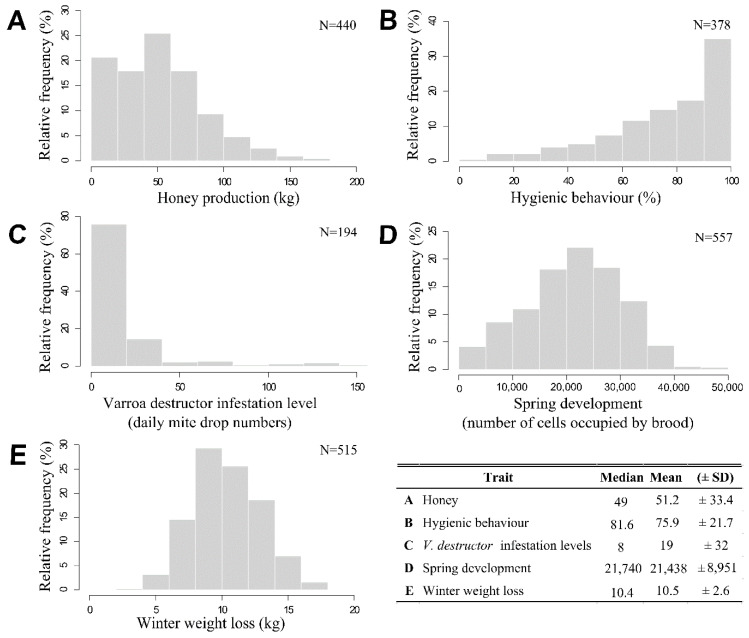
Distribution of phenotypic data between 2010 to 2018 according to traits of interest to our beekeeping industry (prior transformation). (**A**) Distribution of honey production data (kg). (**B**) Distribution of hygienic behavior data (%). (**C**) Distribution of Varroa destructor infestation level data (daily mite drop numbers). (**D**) Distribution of spring development (number of cells occupied by brood). (**E**) Distribution of winter weight loss data (kg) and last a table which include median, mean and standard deviation (±SD) for each traits data between 2010 to 2018.

**Figure 5 insects-11-00587-f005:**
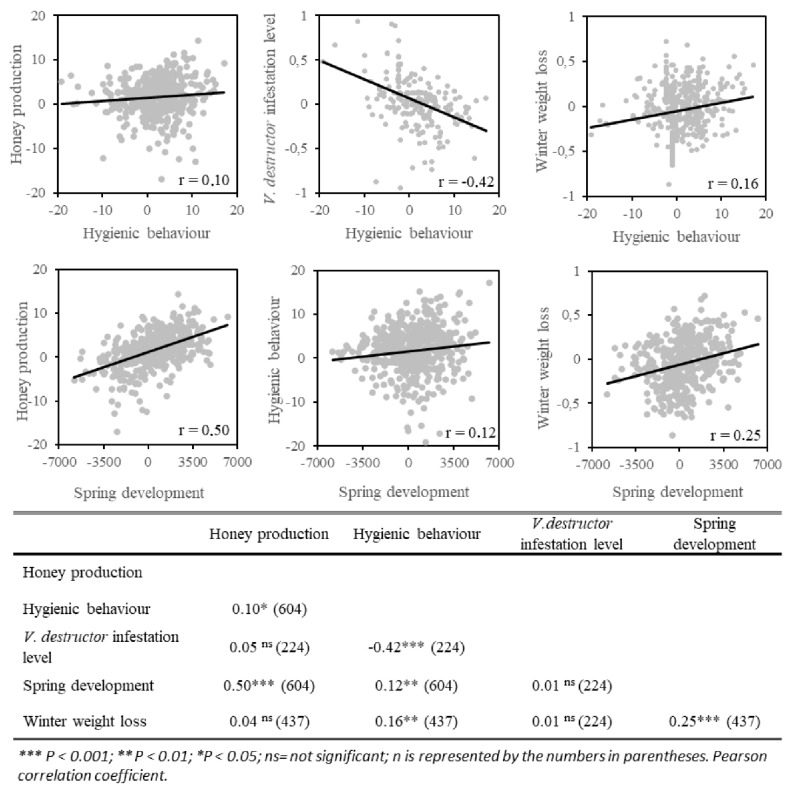
Genetic correlation coefficients among traits of interest to the Canadian beekeeping industry: *Varroa destructor* infestation (daily mite drop numbers), spring development (number of cells occupied by brood), honey production (kg), winter consumption (kg), and hygienic behavior (%).

**Figure 6 insects-11-00587-f006:**
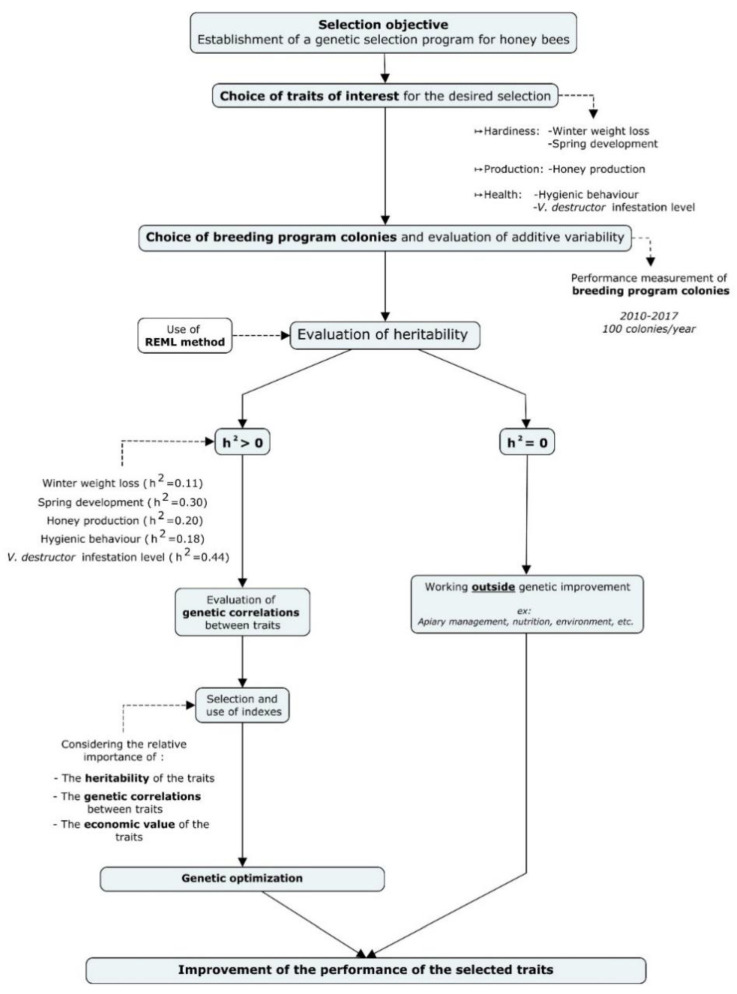
Design of a genetic selection program for honey bees. The first step to establish a genetic selection program is to choose performance traits to improve, as well as the breeding program colony population in which these performance traits will be measured. The second step is to gather performance data on these colonies for several years, and then estimate genetic parameters using the Residual Maximum Likelihood (REML) approach. Estimating the heritability of performance traits shows whether a trait can be selected or not. The last step is to estimate genetic correlations between different traits to better understand the relationship between traits.

**Table 1 insects-11-00587-t001:** Number of colonies and number of queen-producing colonies in each generation in our breeding program.

Generation	Number of Colonies	Number of Queen-Producing Colonies for Breeding of Next Generation
2010	26	7
2011	60	11
2012	38	12
2013	45	13
2014	109	11
2015	144	14
2016	97	9
2017	85	11
Total	604	

**Table 2 insects-11-00587-t002:** Sample size (*n*), variance components (additive variance *V_A_* ± SE and residual variance *V_R_* ± SE) and heritabilities (*h*^2^) with their standard error for traits of interest to the Canadian beekeeping industry.

Trait	*n*	*V_A_* (±SE)	*V_R_* (±SE)	*h*^2^ (±SE)
*Production trait*				
Honey production (kg)	440	74.6 (±50.3)	300.2 (±47.6)	0.20 (±0.13)
*Health traits*				
Hygienic behavior (%)	378	0.32 (±0.24)	1.45 (±0.23)	0.18 (±0.13)
*Varroa destructor* infestation levels (daily mite drop numbers)	194	0.35 (±0.47)	0.43 (±0.41)	0.44 (±0.56)
*Hardiness traits*				
Spring development (number of cells occupied by brood)	557	11,098,058 (±5,514,355)	25,989,661 (±4,731,354)	0.30 (±0.14)
Winter weight loss (kg)	515	0.40 (±0.37)	3.35 (±0.41)	0.11 (±0.09)

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
