# Peer review of "Genetic Parameters of Honey Bee Colonies Traits in a Canadian Selection Program"

_insects, 2020, doi:10.3390/insects11090587_

Round 1
Reviewer 1 Report
Genetic parameters are crucial for the development of breeding programs. While there has been some recent work on genetic parameters for honeybees in temperate and warmer climates, such parameters are rare for colder ones. The size of the dataset, data structure, the statistical methods and the discussion of the present study are only able to provide new insight into this topic to a very limited extent.
Detailed information
L 62-91 A very broad and not very deep introduction to quantitative genetics.
L73: Why do you introduce the composition of the genetic variance into additive and non-additive components, when you do not consider this in your estimates (due to the luck of appropriate data) nor discuss this issue?
L 99-108 Very heterogeneous starting population (undefined European subspecies, Buckfast), which initially leads to a high level of genetic variance, but which is likely not representing the planned (due to size and structure of the) breeding program.
L 112-120 This (as a general problem of parameter estimation in the honey bee) should be described in the introduction and not in the method part.
L126 (and following) The description does not show how the 10 drones producing queens represent the paternal lineage (expected genetic relationship between offspring)
L 154-160 Depending on the (not documented) bee density in the area, the paternal descent of the offspring is doubtful.
L209-212 The simple recording of the number of Varroa mites per colony without taking into account their colony strength will lead to a very imprecise assessment of the correct infestation rate.
L232-235 The mention of MERGE and SORT routines in SAS is no relevant for the statistical method description.
L237-252 The concept of the relationship matrix has not been described. I also think that this is not correctly possible due to the design described. It also remains incomprehensible how your constructed genetic relationship matrix is integrated into the variance component estimation routine of SAS. There is much more suitable software available for estimating genetic variance components (e.g. AS REML etc.)
L 251-252 Pearson correlation coefficients were calculated. This results in phenotypic correlations. How were the (e.g. L287) genetic(?) correlations estimated?
L 254-263 Belongs in the method section (and indeed describes the central problem of variance component estimation in the honey bee).
L287-296, Fig. 5 In my opinion, these are not genetic correlations
Discussion
The discussion differs only slightly from the results section. It should be emphasized that the standard errors of (as expected with the small data set) the heritabilities are very high and therefore precise recommendations are problematic. There is also no discussion of what the results achieved could mean for a breeding program in cooler climates (the real motivation for the study?).
Author Response
Comments and Suggestions for Authors
Genetic parameters are crucial for the development of breeding programs. While there has been some recent work on genetic parameters for honeybees in temperate and warmer climates, such parameters are rare for colder ones. The size of the dataset, data structure, the statistical methods and the discussion of the present study are only able to provide new insight into this topic to a very limited extent.
We have modified our manuscript with the guidance of your specific comments (below)
Detailed information
Point 1: L 62-91 A very broad and not very deep introduction to quantitative genetics.
Response 1: Our introduction gives a concise and broad overview of the basic concepts of quantitative genetics that are essential to understand the scope of our work. A more extensive presentation of quantitative genetics is not our goal here. Furthermore, we give many references that offer the possibility to the lecturer to obtain additional details on quantitative genetics. Please note that the introduction has been changed following a later comment (lines 32-43).
Point 2: L73: Why do you introduce the composition of the genetic variance into additive and non-additive components, when you do not consider this in your estimates (due to the luck of appropriate data) nor discuss this issue?
Response 2: We agree. Sentence has been changed. See lines 89 to 90.
Point 3: L 99-108 Very heterogeneous starting population (undefined European subspecies, Buckfast), which initially leads to a high level of genetic variance, but which is likely not representing the planned (due to size and structure of the) breeding program.
Response 3: This comment is difficult to understand… In Canada, Apis mellifera is not endogenous and we have limited access to specific subspecies. We describe here the origin of our selection baseline honeybee population.
Point 4: L 112-120 This (as a general problem of parameter estimation in the honey bee) should be described in the introduction and not in the method part.
Response 4: We agree. Information has been changed at lines 32 to 48.
Point 5: L126 (and following) The description does not show how the 10 drones producing queens represent the paternal lineage (expected genetic relationship between offspring)
Response 5: We agree, this is confusing. We have removed this information from the Figure 1 at the line 44.
Point 6: L 154-160 Depending on the (not documented) bee density in the area, the paternal descent of the offspring is doubtful.
Response 6: This mating site was chosen based on proximity to our drone producing yard and its minimum distance of 5 km from other unrelated bee yards. We force breed selected males (permanent drone frame) to maximise abundance of our selected drone families in the area. Furthermore, only the queen pedigree is included in our BLUP model (see lines 162 to 163).
Point 7: L209-212 The simple recording of the number of Varroa mites per colony without taking into account their colony strength will lead to a very imprecise assessment of the correct infestation rate.
Response 7: We agree, our explanation was not clear. We added additional information lines 226 to 228. Our BLUP model incorporates May and September varroa infestation rate. Colonies are compared based on the September infestation rate in relation to the initial May infestation rates.
Point 8: L232-235 The mention of MERGE and SORT routines in SAS is no relevant for the statistical method description.
Response 8: We disagree. The estimation of genetic parameters with SAS benefits by the use of these functions. They are essential to sort the databases (Pedigree + Performance) so that they can be integrated by SAS and produce a Kinship Relationship Matrix.
Point 9: L237-252 The concept of the relationship matrix has not been described. I also think that this is not correctly possible due to the design described. It also remains incomprehensible how your constructed genetic relationship matrix is integrated into the variance component estimation routine of SAS. There is much more suitable software available for estimating genetic variance components (e.g. AS REML etc.)
Response 9: We have added more precise information on statistical analysis at the lines 255 to 263.
Point 10: L 251-252 Pearson correlation coefficients were calculated. This results in phenotypic correlations. How were the (e.g. L287) genetic(?) correlations estimated?
Response 10: Additional information has been added at the lines 271 to 273.
Point 11: L 254-263 Belongs in the method section (and indeed describes the central problem of variance component estimation in the honey bee).
Response 11: We agree. Information has been moved in Materials & Methods section, at the lines 192 to 201.
Point 12: L287-296, Fig. 5 In my opinion, these are not genetic correlations
Response 12: This has been explained previously. See lines 271 to 273.
Discussion
Point 13: The discussion differs only slightly from the results section. It should be emphasized that the standard errors of (as expected with the small data set) the heritabilities are very high and therefore precise recommendations are problematic.
Response 13: We agree, but they are nevertheless similar to results obtained in other studies in Canada, (Guarna et al., 2017), USA (Harbo and Harris, 1999) and Europe (Facchini et al., 2019). We have changed the sentence to include this. Lines 353-355.
Point 14: There is also no discussion of what the results achieved could mean for a breeding program in cooler climates (the real motivation for the study?).
Response 14: You will find this information further down in our conclusion. Lines 397 – 402.
Reviewer 2 Report
This study has the potential to help improve honey bee breeding programs, by ensuring selected traits are heritable. However, I’m not sure how much new information has been revealed, as heritability estimates for the traits measured have been calculated by others over the years, particularly in Europe (cited in their literature). I think the novelty of the authors’ approach was to use the Best Linear Unbiased Prediction (BLUP)-Animal Model, used and published by Bienenfeld in 2007 for European bees. As the honey bee colonies in Canada are all descendants of bees from Europe, I’m not sure how results may be expected to differ in Canada; i.e., heritability estimates should be similar in any environment.
I would be very interested to see a modified figure shown in the Supplementary Materials that includes the relative frequency of the measures from colonies in 2018. That is, they show the range in relative frequency for the traits in the original stock, the 7 colonies chosen for the starter stock, and I would like to see another column of figures for the 2018 stock. Did they make progress in selection? As the traits were heritable, and were presumably in a closed population breeding program (daughter queens mated mostly with drones from colonies in selection program), they should be able to show progress in selection.
Line 343: Comparing estimates of hygienic behavior here is not really valid. The authors used the freeze-killed brood test to estimate hygienic behavior. The VSH line from Baton Rouge did not use the same assay: to my knowledge, they introduced mites into cells containing pupae and estimated the removal of mite-infested brood directly. The authors need to compare heritability estimates based on the use of the same assay. There are many estimates based on the freeze-killed brood assay; here is a recent one, but they go back to the 1980’s and probably before that.
Facchini, E., Bijma, P., Pagnacco, G. et al. Hygienic behaviour in honeybees: a comparison of two recording methods and estimation of genetic parameters. Apidologie 50, 163–172 (2019). https://doi.org/10.1007/s13592-018-0627-6
Line 395: In the future, Varroa sensitive hygiene will be measured – how?
As all colonies were treated to control Varroa mites, and no data is given on the effectiveness of the treatments, it is hard to know how to evaluate the measurements of Varroa in colonies, particularly when the assay was the use of sticky boards (which do not estimate mite levels well), and no data was collected in 2015-2017 ( 3 years).
There are a number of English grammar errors, which is understandable as authors are French speaking. I will not go through them line by line, as I hope the journal editors will do so. Here are some that stand out:
- Title should be “Genetic parameters of honey bee colony traits in a …’
- “Rustic” in English means “from the countryside” or “Plain and simple.” I think the authors may be trying to say “locally adapted”?? Line 213 uses the term “Rusticity”, which in addition to meaning “from the countryside” means “graceless or gauche”
Author Response
Comments and Suggestions for Authors
This study has the potential to help improve honey bee breeding programs, by ensuring selected traits are heritable. However, I’m not sure how much new information has been revealed, as heritability estimates for the traits measured have been calculated by others over the years, particularly in Europe (cited in their literature). I think the novelty of the authors’ approach was to use the Best Linear Unbiased Prediction (BLUP)-Animal Model, used and published by Bienenfeld in 2007 for European bees. As the honey bee colonies in Canada are all descendants of bees from Europe, I’m not sure how results may be expected to differ in Canada; i.e., heritability estimates should be similar in any environment.
We have modified our manuscript with the guidance of your specific comments (below).
Point 1: I would be very interested to see a modified figure shown in the Supplementary Materials that includes the relative frequency of the measures from colonies in 2018. That is, they show the range in relative frequency for the traits in the original stock, the 7 colonies chosen for the starter stock, and I would like to see another column of figures for the 2018 stock. Did they make progress in selection? As the traits were heritable, and were presumably in a closed population breeding program (daughter queens mated mostly with drones from colonies in selection program), they should be able to show progress in selection.
Response 1: We have received contradictory suggestions from 2 reviewers and thus have decided to remove this figure 1S. This highlights the emphasis of our manuscript mainly on genetic parameters.
Point 2: Line 343: Comparing estimates of hygienic behavior here is not really valid. The authors used the freeze-killed brood test to estimate hygienic behavior. The VSH line from Baton Rouge did not use the same assay: to my knowledge, they introduced mites into cells containing pupae and estimated the removal of mite-infested brood directly. The authors need to compare heritability estimates based on the use of the same assay. There are many estimates based on the freeze-killed brood assay; here is a recent one, but they go back to the 1980’s and probably before that. Facchini, E., Bijma, P., Pagnacco, G. et al. Hygienic behaviour in honeybees: a comparison of two recording methods and estimation of genetic parameters. Apidologie 50, 163–172 (2019). https://doi.org/10.1007/s13592-018-0627-6
Response 2: We agree that heritability must be measured and compared using of the same essay. We refer to Harbo & Haris1999 that uses the freeze-kill method as we do. We added additional references. See lines 353 to 355.
Point 3: Line 395: In the future, Varroa sensitive hygiene will be measured – how?
Response 3: VSH was removed and reference added for SMR. See line 408.
Point 4: As all colonies were treated to control Varroa mites, and no data is given on the effectiveness of the treatments, it is hard to know how to evaluate the measurements of Varroa in colonies, particularly when the assay was the use of sticky boards (which do not estimate mite levels well), and no data was collected in 2015-2017 (3 years).
Response 4: Additional information was added. See lines 226 to 228.
Point 5: There are a number of English grammar errors, which is understandable as authors are French speaking. I will not go through them line by line, as I hope the journal editors will do so. Here are some that stand out :
- Title should be “Genetic parameters of honey bee colony traits in a …’
- “Rustic” in English means “from the countryside” or “Plain and simple.” I think the authors may be trying to say “locally adapted”?? Line 213 uses the term “Rusticity”, which in addition to meaning “from the countryside” means “graceless or gauche”
Response 5: We have changed title. See lines 2 to 3. “Rustic” has been replaced by “hardy” (see line 62) and “rusticity” has been replaced by “hardiness” (see lines 57, 205, 229, Table 2, Figure 6, 400).
Reviewer 3 Report
In the manuscript authors estimated (co)variances in honey bees as a base for further development of the breeding program. The topic is important since limited publications are available. However, the concept of the paper is not well defined. Authors of the manuscript should decide if they want to present genetic parameters estimation (as title suggest) or breeding program structure. While the estimations of genetic parameters are poorly explained the breeding program is well described. The theory of genetic evaluations explained in introduction is very general without explanation of peculiarities of genetic evaluations in honey bee. Later, material regarding traits’ measurements are explained in details, but data description like number of records per colonies per year and apiary are missing. Specific breeding structure necessary for genetic evaluation is poorly explained in terms how pedigree information was used. Statistical methods used are also not explained appropriately, in terms of model used, methodology and results. It is not clear if transformation of data was done in order to obtain normality in parameters dispersion, and if yes haw it was done. What approach and convergence criteria were used and applied fixed/random effects in the model of evaluation. Generally, it is hard to understand how the evaluation was done. Results can be better explained, while discussion need substantial improvement. Heritability had been explained in introduction and should not be part of discussion. Also comparison of heritability and correlations with results from other researches are expected. Reference list is generally outdated, in particularly to my knowledge there at least 5 new publications from 2019 - 2020 related to estimation (co)variances in honey bees that are not included. I suggest to authors to consider rewriting manuscript in order to reach criteria for publishing.

Author Response
Comments and Suggestions for Authors
In the manuscript authors estimated (co)variances in honey bees as a base for further development of the breeding program. The topic is important since limited publications are available. However, the concept of the paper is not well defined. Authors of the manuscript should decide if they want to present genetic parameters estimation (as title suggest) or breeding program structure. While the estimations of genetic parameters are poorly explained the breeding program is well described. The theory of genetic evaluations explained in introduction is very general without explanation of peculiarities of genetic evaluations in honey bee. Later, material regarding traits’ measurements are explained in details, but data description like number of records per colonies per year and apiary are missing. Specific breeding structure necessary for genetic evaluation is poorly explained in terms how pedigree information was used. Statistical methods used are also not explained appropriately, in terms of model used, methodology and results. It is not clear if transformation of data was done in order to obtain normality in parameters dispersion, and if yes haw it was done. What approach and convergence criteria were used and applied fixed/random effects in the model of evaluation. Generally, it is hard to understand how the evaluation was done. Results can be better explained, while discussion need substantial improvement. Heritability had been explained in introduction and should not be part of discussion. Also comparison of heritability and correlations with results from other researches are expected. Reference list is generally outdated, in particularly to my knowledge there at least 5 new publications from 2019 - 2020 related to estimation (co)variances in honey bees that are not included. I suggest to authors to consider rewriting manuscript in order to reach criteria for publishing.
We have modified our manuscript with the guidance of your specific comments (below).
In particular :
- TITLE:
Point 1: Authors should decide either to describe Canadian breeding program or to be focused on genetic parameters estimation. Also, norther selection program is not appropriate, Canadian selection program can be better.
Response 1: We have changed title. See lines 2 to 3.
- INTRODUCTION:
Point 2: L71: “N to the identified effects of the environment (fixed effects)” omit fixed effects, they can be treated as random too in modeling. It is depending to modeling choice.
Response 2: This is the basic quantitative genetics equation. (Minvielle, F. Principes d’amélioration génétique des animaux domestiques, 1st ed. ; Les presses de l’université Laval : Québec, Canada, 1990 ; pp. 15-68 ; Falconer, D.S.; Mackay, T.F.C. Introduction to Quantitative Genetics, 4th ed.; P. Longman ed.; Longman: Burnt Mill, England, 1996; pp. 135-165 ; Néron, F.; Guéguen, R. Petit précis d’élevage, 1st ed.; France Agricole ed.; Agriproduction : Paris, France, 2018 ; pp. 310-443.)
Point 3: L72: “E to the residual effects of the environment (random effects)”. Residual is not only due to residual effects of the environment, hence delete “effects of the environment”
Response 3: This is the basic quantitative genetics equation. (Minvielle, F. Principes d’amélioration génétique des animaux domestiques, 1st ed. ; Les presses de l’université Laval : Québec, Canada, 1990 ; pp. 15-68 ; Falconer, D.S.; Mackay, T.F.C. Introduction to Quantitative Genetics, 4th ed.; P. Longman ed.; Longman: Burnt Mill, England, 1996; pp. 135-165 ; Néron, F.; Guéguen, R. Petit précis d’élevage, 1st ed.; France Agricole ed.; Agriproduction : Paris, France, 2018 ; pp. 310-443.)
- MATERIAL & METHODS:
Point 4: L128: How “entire group of fathers …” were defined in pedigree file? It is important for later use in relationship matrix construction. Explain!
Response 4: We agree, additional information was added. See lines 254 to 263.
Point 5: L131‐132: What was consider as performance in the evaluation, queen of the colony or colony itself (queen + worker bees)? Explain!
Response 5: See L207: 2.3. Traits measured to evaluate colony performance. We also added additional information, see lines 131 to 133.
Point 6: L135: “our pedigree database contains the performance of 604 colonies …” In pedigree data base we expect to see number of individuals and their ancestors, average number of generation per individual, inbreeding…. It is important to understand the quality of pedigree data for genetic evaluation. If genetic ties are not sufficient then evaluation will result in low reliability. Performance information are part of performance data. Redefine sentence!
Response 6: We agree and we added TABLE 1 line 140.
Point 7: L156: “83‐93% of mating is performed by the selected males.” Meaning that fathers of queens are not 100% know. How this is reflecting on pedigree (relationship) information is expected to be explained.
Response 7: Information has been added. See lines 162 to 165.
Point 8: L226: Refers to performance database, additional argument for confusion in L135.
Response 8: This information can be found at 2.3 Traits measured to evaluate colony performance.
Point 9: L236: It is good that “All dependent variables were tested for normality “, but it is not clear for which traits transformation was used and how it changes distribution. Explain!
Response 9: Information has been added. See lines 253 to 254.
Point 10: L241‐242: “Variance components of phenotype traits were estimated with the Best Linear Unbiased Prediction‐Animal Model (linear mixed model) using the Mixed procedure.” makes the biggest confusion. BLUP is a method that is used for breeding value estimation, while (co)variance estimates are usually obtained by other approaches like REML. (Co)variances should be priory estimated and used in BLUP evaluations. Hence it is not clear at all how (co)variances are estimated. Also, model is missing? What systematic effects and in which form were included in evaluation? The most critical part of the manuscript in order to understand results!
Response 10: We agree. Information has been added. See lines 254 to 263.
Point 11: L243‐247: not clear at all how standard errors were calculated. In sentence “SAS software does not deliver standard errors directly in the outputs, so they were estimated from the variances of the heritability’s and standard errors of each trait with….” needs deep reformulation, look like a loop.
Response 11: We agree. Sentence has been changed. See lines 265 to 267.
Point 12: L248: what is var(Vadditive) variance of additive variance? Explain!
Response 12: Additional explanation above and reference is given. See lines 274 to 276.
Point 13: L251: Explain how genetic covariances are obtained? Genetic correlations require genetic covariances, usually estimated by multiple trait models.
Response 13: See above for explanation. See lines 265 to 267.
- RESULTS:
Point 14: L254‐257: should be part of MM not in Results.
Response 14: We agree. Information has been moved in Materials & Methods section, See 2.1 Pedigree determination lines at the lines 192 to 201.
Point 15: L258‐260: in MM. Now I understood that males were not considered in pedigree (relationship). So all explanations in MM related to drone in pedigree structure need to be rewrite.
Response 15: Additional information has been added. See lines 161 to 165.
Point 16: L261‐262: is discussion not results.
Response 16: We agree. Information has been moved in Discussion section, at the lines 364 to 366.
Point 17: L271‐277: Distributions (Fig.4) are prior or after transformation? Explain!
Response 17: Additional information has been added. See line 277.
Point 18: L278: Results in table should be considered with percussion, due to unclear methodology used, and very high SE of variances. Also it is difficult to discuss results with SE that are larger than the value itself (for Varroa VA=0.35+/‐0.47).
Response 18: We agree, that heritability must be done on the use of the same essay. We refer to Harbo 1999 that uses the freeze-kill method as we do. We added additional references. See lines 353 to 355.
Point 19: L287‐295: Not clear how genetic correlations were calculated, perhaps one can understand how to take results when methodology would be better explained. The results can be different!
Response 19: Additional information added. See lines 271 to 273.
- DISCUSSION:
If authors decide to discuss for breeding program than comments below are not appropriate.
Point 20: L303‐305: Variance cannot be estimated with BLUP – Animal Model, authors should be correct in terminology use (also in Figure 6).
Response 20: In text (2.4 Statistical analysis, see lines 259 to 263) and Figure 6 (line 325) we explain how we use BLUP-Animal Model.
Point 21: L306‐312: is mostly explained in introduction, so there is no need for it.
Response 21: We have not changed this. We consider that this information is important.
Point 22: L 306 & L315: Figure 6, I see no point of it and also discussion about it. The title is referring for genetic parameters.
Response 22: We have not changed this. We consider that this information is important and instructive.
Point 23: L322‐330: very general discussion
Response 23: No comment
Point 24: L331‐351: in discussion compare results with latest papers as:
Response 24:
Gertje Eta Leony Petersen, Fennessy P.F., Amer P.R., Dearden P.K. (2020)
Designing and implementing a genetic improvement program in commercial
beekeeping operations, Journal of Apicultural Research, DOI:
10.1080/00218839.2020.1715583
→ Very interesting review manuscript
Guichard, M., Neuditschko, M., Soland, G. et al. (2020). Estimates of genetic
parameters for production, behaviour, and health traits in two Swiss honey
bee populations. Apidologie https://doi.org/10.1007/s13592‐020‐00768‐z.
→ Reference was added line 346.
Boecking O., Bienefeld K., Drescher W. (2020) Heritability of the Varroa specific
hygienic behaviour in honey bees (Hymenoptera: Apidae). J. Animal
Breed. Genet. 117, 417±424
→ Unfortunately, we do not agree to compare results of the pin test vs freeze killed brood test.
Andonov, S., Costa, C., Uzunov, A. et al. Modeling honey yield, defensive and
swarming behaviors of Italian honey bees (Apis mellifera ligustica) using
linear‐threshold approaches. BMC Genet 20, 78 (2019).
https://doi.org/10.1186/s12863‐019‐0776‐2
→ Results and reference were added lines 345 to 346.
Facchini E., Bijmap., Pagnacco G., Rizzi R., Brascamp EW. (2019) Hygienic
behaviour in honeybees: a comparison of two recording methods and
estimation of genetic parameters. Apidologie 50:163–172. DOI:
10.1007/s13592‐018‐0627‐6
→ Results and reference were added lines 353 to 355.
Brascamp EW, Willam A, Boigenzahn C, Bijma P, Veerkamp RF. (2018)
Correction to: Heritabilities and genetic correlations for honey yield,
gentleness, calmness and swarming behaviour in Austrian honey bees.
Apidologie. 49:462–3
→ Results and reference were added line 346.
Point 25: 6. CONCLUSION and ABSTRACT should be reconsidered after clarification of methodology used.
Response 25: This has been done.
Point 26: 7. KEY WORDS reduce to 5.
Response 26: Only 5 key words have been keep. See lines 24.
Point 27: 8. REFFERENCES: update the list.
Response 27: Reference has been updated.
Point 28: 9. Supplement Figure 1 should be omitted
Response 28: Figure 1S has been removed
Round 2
Reviewer 1 Report
The changes made by the authors only slightly improve the manuscript. The new explanations on the methodology of parameter estimation make the problems even more obvious. The method of heritability estimation is not state of the art, as the SAS routine used cannot generate a relationship matrix that suits the particular features of the honey bee. The procedure used cannot be called BLUP.
The method of estimating genetic correlation is also wrong. Pearson's correlation coefficients between the estimated breeding values are not genetic correlations.
In addition to the methodological problems in parameter estimation, the size of the data set (and thus the level of standard errors) does not provide a solid basis for developing strategies for breeding programs.
Author Response
Point 1: The changes made by the authors only slightly improve the manuscript. The new explanations on the methodology of parameter estimation make the problems even more obvious. The method of heritability estimation is not state of the art, as the SAS routine used cannot generate a relationship matrix that suits the particular features of the honey bee. The procedure used cannot be called BLUP.
Response 1: We agree, additional information was added. See lines 253 to 282.
Point 2: The method of estimating genetic correlation is also wrong. Pearson's correlation coefficients between the estimated breeding values are not genetic correlations.
Response 2: We agree, additional information was added. See lines 283 to 285.
Point 3: In addition to the methodological problems in parameter estimation, the size of the data set (and thus the level of standard errors) does not provide a solid basis for developing strategies for breeding programs.
Response 3: We have changed our discussion accordingly to explain these facts. See lines 377 to 381.
Reviewer 2 Report
The authors have addressed all of my comments.
I disagree with removing Fig 1S and still would have liked to see if the made progress in selection, but hopefully this will be a topic for a future manuscript.
Author Response
Thank you for your constructive comments and corrections which have provided a real added value to our manuscript. We are currently working on a future manuscript on the genetic progression of traits in which we will present results related to Figure S1.
Sincerely,
Ségolène Maucourt
Reviewer 3 Report
In the manuscript authors estimated (co)variances in honey bees as a base for further development of the breeding program. Second version of the manuscript has been slightly improved, but still I am not convinced that it meets criteria for publication Authors are not using proper terminology, genetic variances and heritabilities are estimated not measured nor determined. BLUP as method can’t estimate variances, but breeding values. SAS procedure used for BLUP has in background variance components’ estimator (default is REML, but there are options for other approaches, too), but authors did not explain that. SAS MIXED procedure implies combining fixed and random effects. Since model has not been present it is not clear at all what effects were used and how were treated in the evaluation. Most of the substantial comments given before have not been considered or sufficiently explained (yellow highlighted). Hence, I suggest to editor not to accept the paper, or ask authors to rewrite manuscript with correct explanation of statistical methods used.

Author Response
Genetic parameters of honey bee colonies trait’s in a Canadian selection program
General comments:
In the manuscript authors estimated (co)variances in honey bees as a base for further development of the breeding program. Second version of the manuscript has been slightly improved, but still I am not convinced that it meets criteria for publication Authors are not using proper terminology, genetic variances and heritabilities are estimated not measured nor determined. BLUP as method can’t estimate variances, but breeding values. SAS procedure used for BLUP has in background variance components’ estimator (default is REML, but there are options for other approaches, too), but authors did not explain that. SAS MIXED procedure implies combining fixed and random effects. Since model has not been present it is not clear at all what effects were used and how were treated in the evaluation. Most of the substantial comments given before have not been considered or sufficiently explained (yellow highlighted). Hence, I suggest to editor not to accept the paper, or ask authors to rewrite manuscript with correct explanation of statistical methods used.
- INTRODUCTION:
Point 1: L71, now L85: “N to the identified effects of the environment (fixed effects)” omit fixed effects, they can be treated as random too in modeling. It is depending to modeling choice. It has not been accepted.
Response 1: We agree, additional information was added. See lines 85 to 86.
Point 2: L72, now L86: “E to the residual effects of the environment (random effects)”. Residual is not only due to residual effects of the environment, hence delete “effects of the environment” It has not been accepted.
Response 2: We agree, the sentence has been modified. See line 86.
Point 3: L109: Genetic parameters are estimated, not measured. Authors demonstrated lack of knowledge for methods applied and terminology used.
Response 3: We agree, the term was changed. See line 109.
MATERIAL & METHODS:
Point 4: L128, now L128: How “entire group of fathers …” were defined in pedigree file? It is important for later use in relationship matrix construction. Explain! It has not been improved.
Response 4: We agree, additional information was added. See lines 126 to 129.
Point 5: L135, now L136: “our pedigree database contains the performance of 604 colonies …” In pedigree data base we expect to see number of individuals and their ancestors, average number of generation per individual, inbreeding…. It is important to understand the quality of pedigree data for genetic evaluation. If genetic ties are not sufficient then evaluation will result in low reliability. Performance information are part of performance data. Redefine sentence! It has not been sufficiently improved. By adding table only partly has been responded, and still one can’t get information for number of individuals with know /unknown ancestors, average number of generation per individual, inbreeding.
Response 5: We agree, additional information was added. See lines 138 to 156.
Point 6: L158: “83-93% of mating is performed by the selected males.” Meaning that fathers of queens are not 100% know. How this is reflecting on pedigree (relationship) information is expected to be explained. Explanation was added but still not convenient what was done and how, particularly with sentence:
L164-165:” Unfortunately, there is no guarantee that our queens are 100% fertilized by these selected males but we are confident that most of our queens are fertilized by them [38].”
Response 6: Drones-producing colonies are indicated in our pedigree file as shown in lines 130 to 134. We have also indicated that the genetic information on the male side was not considered in the statistical model for our analyses, see lines 168 to 169, 202 to 207, 377 to 381, and 386 to 390. In our opinion, this information is sufficient.
Point 7: L192-194: Why data form problematic colonies were removed, in particularly form pedigree file? Explain!
Response 7: There is a confusion with the line number. We think that the following lines 198 to 201 have the missing information.
Point 8: L199-200: Sentence should be in discussion!
Response 8: Reviewer 1 had requested that this passage be placed in the material and method section.
Point 9: L236: now L253-254: It is good that “All dependent variables were tested for normality “, but it is not clear for which traits transformation was used and how it changes distribution. Partly explained, one expects more explicit explanation by presenting distributing before and after transformation.
Response 9: Traits that were transformed to meet normality are all identified in lines 257 to 260. Statistical analyses were done on transformed data. We decided to present distribution of phenotypic data prior transformation for ease of comprehension and we made it clear in caption of Figure 4.
Point 10: L241-242: now L 259-260: “Variance components of phenotype traits were estimated with the Best Linear Unbiased Prediction-Animal Model (linear mixed model) using the MIXED procedure.” makes the biggest confusion. BLUP is a method that is used for breeding value estimation, while (co)variance estimates are usually obtained by other approaches like REML. (Co)variances should be priory estimated and used in BLUP evaluations. Hence it is not clear at all how (co)variances are estimated. Also, model is missing? What systematic effects and in which form were included in evaluation? The most critical part of the manuscript in order to understand results! Authors did not respond to comments addressed to them.
Response 10: We have realized that important information on our model was lacking. We have rewritten the paragraph line 254 to 285 and hope that this is sufficient.
- RESULTS:
Point 11: L278: Results in table should be considered with percussion, due to unclear methodology used, and very high SE of variances. Also it is difficult to discuss results with SE that are larger than the value itself (for Varroa VA=0.35+/-0.47). It has not been improved.
Response 11: A proposed explanation of the high SE of varroa infestation level of heritability has been added in discussion line 379 to 381. High levels of SE have also been estimated in other research (h2= 0.65±0.61 in Harbo & Harris 1999, Heritability in Honey Bees (Hymenoptera: Apidae) of Characteristics Associated with Resistance to Varroa jacobsoni (Mesostigmata: Varroidae) ; h2=0.37 ±0.25 in Facchini & al. 2019, Hygienic behaviour in honeybees: a comparison of two recording methods and estimation of genetic parameters ; h2=0.09±0.06 in Guichard & al. 2020, Estimates of genetic parameters for production, behaviour, and health traits in two Swiss honey bee populations).
Point 12: L287-295: Not clear how genetic correlations were calculated, perhaps one can understand how to take results when methodology would be better explained. The results can be different! It has not been improved.
Response 12: Explanation added line 283 to 285.
- DISCUSSION:
If authors decide to discuss for breeding program than comments below are not appropriate.
Point 13: L303-305: now L315-316: Variance cannot be estimated with BLUP – Animal Model, authors should be correct in terminology use. It has not been improved.
Response 13: We changed the terminology line 328
Point 14: L306-312: is mostly explained in introduction, so there is no need for it. It has not been improved.
Response 14: We respectfully disagree with the reviewer as we believe these sentences are important to start our discussion.
Point 15: L 306 & L315: Figure 6, I see no point of it and also discussion about it. The title is referring for genetic parameters. It has not been improved.
Response 15: We respectfully disagree with the reviewer as we believe Figure 6 adds complementary information for the readers to understand why it is important to estimate genetic parameters of interest traits for selection when considering starting a honey bee genetic selection program.
Point 16: L322-330: very general discussion It has not been improved.
Response 16: We did not change the Discussion section as we believe it is the section to provide a broad overview of the work. However, information has been added due to several comments.
Point 17: CONCLUSION and ABSTRACT should be reconsidered after clarification of methodology used.
Response 17: This has been done.
Round 3
Reviewer 3 Report
Manuscript has been improved accordingly, small improvements should be incorporated:
L264-265: delete Best Linear Unbiased Prediction, REML - Animal model only.
L328 and 343: replace REML method with REML approach
Author Response
Thank you for your helpful comments. We have made the requested corrections.
Point 1: L264-265: delete Best Linear Unbiased Prediction, REML - Animal model only.
Response 1: We changed the sentence. See lines 264 to 265.
Point 2: L328 and 343: replace REML method with REML approach
Response 2: We changed method for approach. See lines 328 and 343.